# Balance between BDNF and Semaphorins gates the innervation of the mammary gland

**Hadas Sar Shalom, Ron Goldner, Yarden Golan-Vaishenker, Avraham Yaron***

Department of Biomolecular Sciences, Weizmann Institute of Science, Rehovot, Israel

**Abstract** The innervation of the mammary gland is controlled by brain-derived neurotrophic factor (BDNF), and sexually dimorphic sequestering of BDNF by the truncated form of TrkB (TrkB.T1) directs male-specific axonal pruning in mice. It is unknown whether other cues modulate these processes. We detected specific, non-dimorphic, expression of Semaphorin family members in the mouse mammary gland, which signal through PlexinA4. PlexinA4 deletion in both female and male embryos caused developmental hyperinnervation of the gland, which could be reduced by genetic co-reduction of BDNF. Moreover, in males, PlexinA4 ablation delayed axonal pruning, independently of the initial levels of innervation. In support of this, in vitro reduction of BDNF induced axonal hypersensitivity to PlexinA4 signaling. Overall, our study shows that precise sensory innervation of the mammary gland is regulated by the balance between trophic and repulsive signaling. Upon inhibition of trophic signaling, these repulsive factors may promote axonal pruning.
DOI: https://doi.org/10.7554/eLife.41162.001

## Introduction

During development of the peripheral nervous system, specific neuronal connections are being established to ensure its proper function. Sensory neurons of the dorsal root ganglia (DRG) initially extend their axons toward their targets by responding to attractive and repulsive cues expressed in the extracellular environment along their trajectory (*Kolodkin and Tessier-Lavigne, 2011*). Once axons reach the targets, limiting amounts of neurotrophic factors control target innervation by regulating axonal growth, neuronal cell death, and pruning (*Huang and Reichardt, 2001*).

Brain-derived neurotrophic factor (BDNF) acts as a survival factor for specific subsets of sensory neurons during development (*Kirstein and Fariñas, 2002*), and regulates mechanosensation (*Carroll et al., 1998*). During early development of the mammary gland, BDNF is required to establish and maintain sensory innervation in both female and male embryos. In males only, at later stages of development, sequestering of BDNF by a truncated form of TrkB (TrkB.T1) results in elimination of the axons innervating the gland. This elimination is not blocked in the Bax knockout (KO), suggesting that it occurs through axonal pruning rather than cell death (*Liu et al., 2012*). As a result, a sexually dimorphic pattern of mammary gland sensory innervations is formed. Whether other cues operate in concert with BDNF, both in females and males, is unknown.

The Semaphorins belong to a large family of secreted and membrane-bound guidance cues. The 27 family members are divided into 8 subclasses according to structural homology (*Pasterkamp, 2012*). Initially characterized for their importance in axonal guidance during development, Semaphorins are now known to be involved in multiple developmental processes including axonal fasciculation, pruning, dendritic guidance, and modulation of the immune system (*Pasterkamp and Kolodkin, 2003*; *Yazdani and Terman, 2006*; *Tang et al., 2007*; *Zhou et al., 2008*). Two receptor families have been implicated in mediating Semaphorin family functions: Plexins and Neuropilins

***For correspondence:**
avraham.yaron@weizmann.ac.il

**Competing interests:** The authors declare that no competing interests exist.

**eLife digest** Almost every action an animal can perform in its life will rely on its nervous system being wired correctly. Before the animal is born, nerve cells next to the spinal cord send out long fibers – called axons – to connect with different parts of its body. These nerves will help relay sensations to the brain. The ends of the nerve fibers follow trails of signals that guide them to the correct targets. When the axon arrives in the right place, it can receive further signals called neurotrophic factors. These signals keep the axon alive, but they are in short supply. Not every axon will receive the signal, and, without it, the nerve fiber dies back. This phenomenon, known as pruning, helps to make sure that nervous system does not form more connections than it needs.

In mammals, the mammary gland is an example of a part of the body where nerve endings are pruned during development. Axons that connect to this milk-producing gland depend on a neurotrophic factor called BDNF to survive, and BDNF is controlled differently in males and females. In male mammals, another protein grabs hold of the BDNF signal and hides it away before development is complete. After this happens, the axons start to die, however it was not clear if other signals are also involved.

Sar Shalom et al. have now examined if proteins called Semaphorins – which guide axons to their target locations and influence pruning too – also control how many nerves end up connected to the mammary gland. The experiments used nerve cells grown in the laboratory and genetically modified mice, and suggested that the nerves in the mammary gland would only develop correctly if the BDNF and Semaphorins signals were properly balanced.

When the lab-grown nerve cells encountered Semaphorins, their growing axons collapsed. Yet unlike for BDNF, the levels of Semaphorins were the same in male and female mice. Further experiments showed that if a protein receptor that detects the Semaphorins was deleted, the nerve cells stopped responding to the signal, and their axons did not collapse. In mice lacking this receptor, both sexes ended up with more axons in their mammary glands. Too many axons grew in the female mice, while the pruning of excess axons was delayed in the males. Reducing the levels of BDNF in these mice helped to return axon growth to normal. Together, these findings suggest that a balance between the BDNF and the Semaphorins sets the correct number of nerves. They also suggest that once the BNDF signal is removed during the normal development of males, it is the Semaphorins that help the axons in the mammary gland to be pruned.

Lastly, neurotrophic factors and Semaphorins are not just important during development; indeed cells make them well into adulthood. Altered patterns of these signals in mature animals could change the shapes of nerve networks. As such, future work may help scientists to understand why tissues can become too sensitive or lose sensation.

DOI: https://doi.org/10.7554/eLife.41162.002

(*Fujisawa and Kitsukawa, 1998*). Most class 3 Semaphorins bind to Neuropilins, which associate with type A Plexin family members to transduce their signals through the plasma membrane. Other members of the Semaphorin family bind and signal directly through Plexin receptors (*Yoshida, 2012*). Moreover, previous in vitro and overexpression studies demonstrated a crosstalk between class 3 Semaphorins and neurotrophins (*Dontchev and Letourneau, 2002*; *Atwal et al., 2003*; *Tang et al., 2004*; *Ben-Zvi et al., 2006*; *Tang et al., 2007*; *Ben-Zvi et al., 2008*).

Here we show that opposing effects of neurotrophic (BDNF) and repulsive (Semaphorins) factors regulate the sensory innervation of the mammary gland, and that the repulsive factors may promote axonal pruning once trophic signaling is inhibited.

## Results

### TrkB but not Semaphorins exhibit sexually dimorphic expression in the mammary gland

The sexually dimorphic innervation of the mammary gland at late E13 mouse embryos is a result of differential expression of TrkB.T1 in mesenchymal cells surrounding the mammary epithelium. In male embryos, expression of TrkB.T1 results in sequestering of BDNF, which leads to pruning of the

sensory fibers innervating the mammary gland (*Liu et al., 2012*). In agreement, we have detected this expression pattern of TrkB.T1 using an antibody that recognizes the TrkB extracellular domain (TrkB[ECD]) (*Figure 1B* arrowheads). In addition, this staining revealed innervation of the gland by TrkB-positive axons in female embryos and reduced innervation in male embryos (*Figure 1A,B* arrows). In order to find additional factors that regulate the innervation, we performed an in-situ hybridization screen to detect the mRNA that encodes the different Semaphorins (*Figure 1C–H* and *Figure 1—figure supplement 1*). Our screen detected clear expression of the secreted Semaphorins, *Semaphorin3d* (*Sema3d*) and *Semaphorin3f* (*Sema3f*), in the epithelial cells of the gland (*Figure 1C–F*). Less profound expression of the membrane-bound *Semaphorin6a* (*Sema6a*) was detected as well (*Figure 1G,H*). There was no signal by the sense control probes (*Figure 1—figure supplement 2*). To confirm expression of *Sema6a*, we took advantage of the *Sema6a* genetrap line that allows detection of Sema6A-expressing cells by X-gal staining (*Leighton et al., 2001*). Whole mount staining showed similar epithelial expression of Sema6A in female and male embryo mammary glands (*Figure 1I,J*). Unlike TrkB.T1 expression, the expression pattern of the three Semaphorins was identical between female and male embryos.

## Sema6A and Sema3D induce growth cone collapse of BDNF-responsive neurons in a PlexinA4-dependent manner

To determine whether BDNF-responsive neurons that innervate the mammary gland respond to Semaphorins expressed in the epithelial cells of the gland, we used an in vitro growth cone collapse assay. DRG explants from early E13 embryos were grown in the presence of BDNF for 48 hr, and then soluble Semaphorins were added to the media for 30 min, after which the axonal growth cones were detected by staining for actin using Phalloidin-Rhodamine. Under basal conditions, between 20% and 40% of the growth cones exhibited a collapsed morphology. Both Sema6A and Sema3D increased this fraction to about 60% of the growth cones. In contrast, Sema3F had no effect (*Figure 2A–F*). The PlexinA4 receptor can propagate repulsive activities of both secreted and

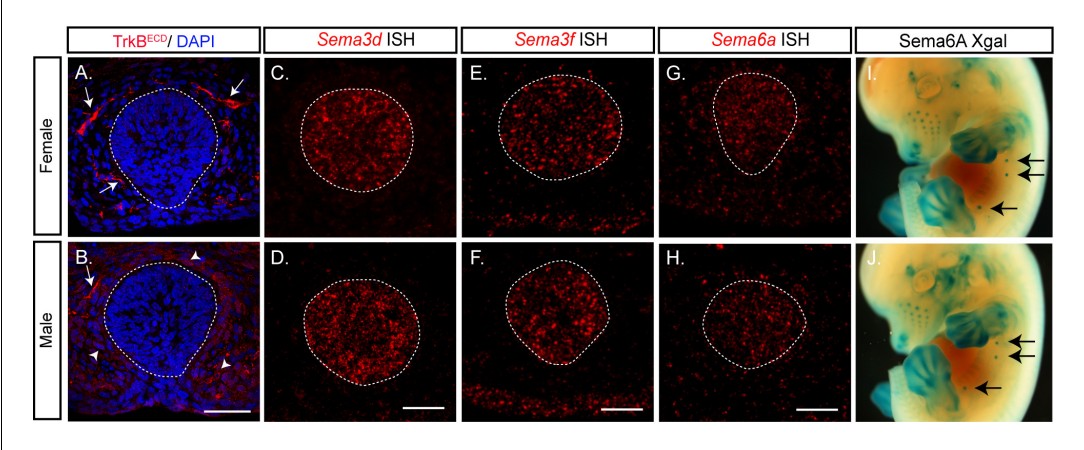

**Figure 1.** TrkB displays sexually dimorphic expression in the mammary gland while members of the Semaphorin family display a non-dimorphic expression. (A, B) Mammary gland sections of late E13 female and male embryos were stained with TrkB[ECD] antibody and counterstained with DAPI for visualization of the mammary gland structure. Arrows indicate TrkB-expressing sensory axons innervating the mammary gland; arrowheads point to the male-specific expression of TrkB in the mesenchymal cells surrounding the gland. (C–H) In-situ hybridization of mammary gland sections of late E13 female (C, E, G) and male (D, F, H) embryos using specific digoxigenin (DIG)-labeled RNA probes for *Sema3d*, *Sema3f* and *Sema6f*, as indicated. The mammary epithelium is marked by a circle according to the DAPI staining. Scale bar: 50 µm. (I, J) Whole mount X-gal staining of late E12 Sema6A Het female and male mutant embryos. Glands 2–4 are marked by arrows, glands 1 and 5 are hidden by the limbs.

DOI: https://doi.org/10.7554/eLife.41162.003

The following figure supplements are available for figure 1:

**Figure supplement 1.** In-Situ hybridization screen for Semaphorins expression in the mammary gland.
DOI: https://doi.org/10.7554/eLife.41162.004

**Figure supplement 2.** Sense controls of *Sema3d*, *Sema3f* and *Sema6a*.
DOI: https://doi.org/10.7554/eLife.41162.005

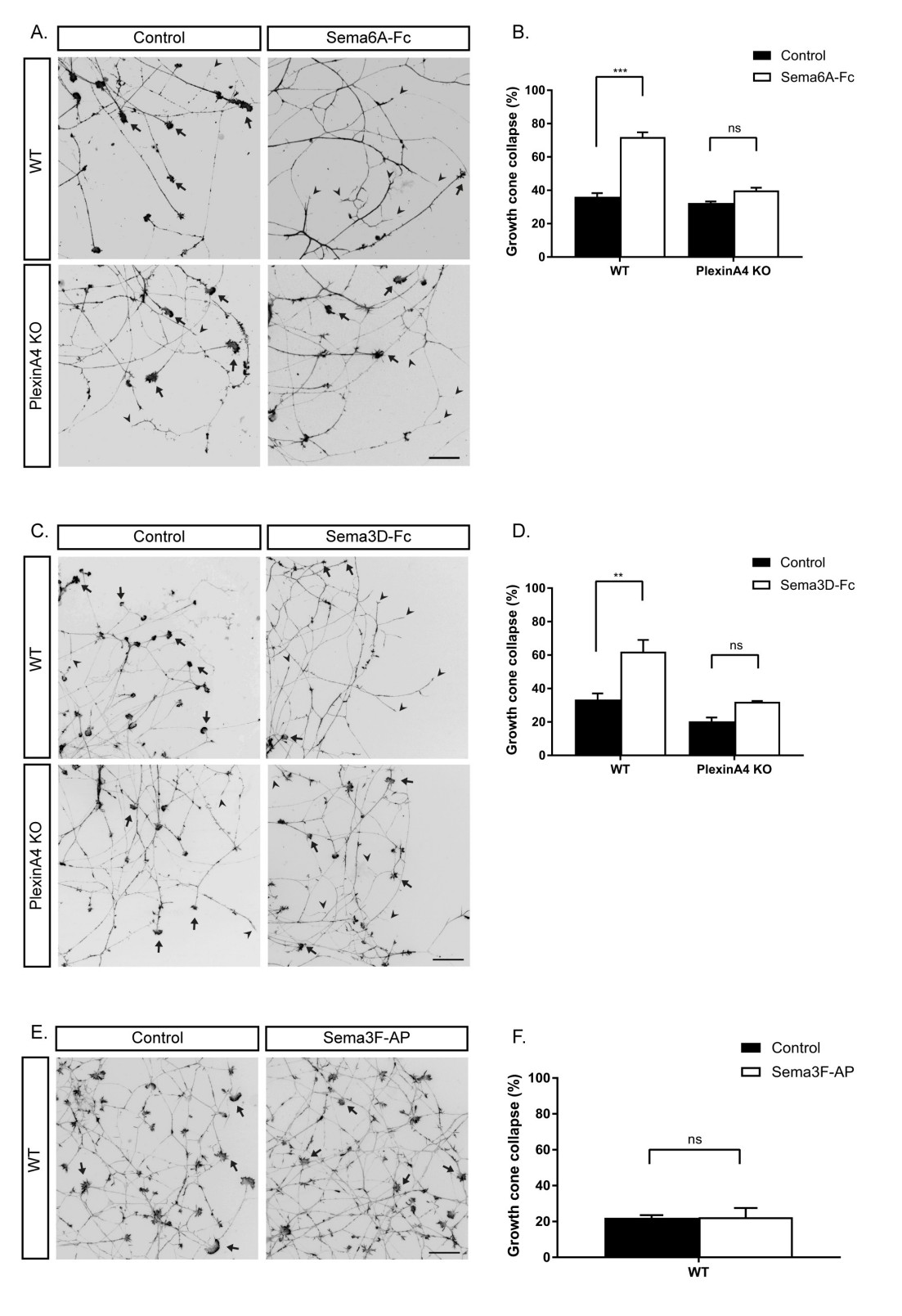

**Figure 2.** In vitro growth cone collapse of BDNF-responsive neurons by Sema6A and Sema3D is PlexinA4 dependent. (**A, C, E**) DRG explants from early E13 WT or PlexinA4 KO embryos were grown for 48 hr, treated with 1 ng/µl soluble Sema6A-Fc (**A**), 0.02 ng/µl soluble Sema3D-Fc (**C**), 1 nM of Sema3F-AP (**E**), or control media for 30 min, fixed, and stained with Phalloidin-Rhodamine to visualize growth cones. Black arrowheads indicate collapsed growth cones and arrows indicate intact growth cones. Scale bar: 50 µm. (**B, D, F**) Quantification of collapsed growth cones as percentage of total,

*Figure 2 continued on next page*

Figure 2 continued

means ±SEM of more than 1000 growth cones obtained from three independent experiments. (B) WT: control vs Sema6A-Fc, p<0.0001; PlexinA4 KO: control vs Sema6A-Fc, p=0.110307. (D) WT: control vs Sema3D-Fc, p=0.0053; PlexinA4 KO: control vs Sema3D-Fc, p=0.2666. (F) Control vs Sema3F-AP, p=0.954. ***p<0.001, **p<0.01, ns - non-significant, two-way ANOVA with *post hoc* analysis.

DOI: https://doi.org/10.7554/eLife.41162.006

The following source data and figure supplement are available for figure 2:

**Source data 1.** Percentage of collapsed growth cones in WT and PlexinA4 KO.

DOI: https://doi.org/10.7554/eLife.41162.008

**Figure supplement 1.** Sema6A binds solely to PlexinA4 in BDNF-responsive sensory neurons.

DOI: https://doi.org/10.7554/eLife.41162.007

transmembrane Semaphorins (*Suto et al., 2005*; *Tran et al., 2007*; *Yoshida, 2012*). Sema6A can bind and signal directly through both PlexinA4 or PlexinA2 (*Suto et al., 2005*; *Suto et al., 2007*). PlexinA4 also acts as the signaling co-receptor for Sema3A in a complex with Neuropilin1 (Nrp1) (*Suto et al., 2005*; *Yaron et al., 2005*; *Tran et al., 2007*; *Yoshida, 2012*). While Sema3D was shown to bind Nrp1 (*Yazdani and Terman, 2006*), the specific Plexin through which it signals was not reported. Therefore, we tested if Sema3D and Sema6A signal through PlexinA4 in BDNF-responsive neurons by performing a collapse assay using DRGs of PlexinA4 KO mice (*Yaron et al., 2005*). Interestingly, we detected complete loss of response both to Sema6A and Sema3D in these neurons (*Figure 2A–D*), demonstrating that PlexinA4 serves as the signaling receptor for both Sema6A and Sema3D. Since Sema6A can also bind and signal through PlexinA2 (*Suto et al., 2005*; *Suto et al., 2007*), we tested if PlexinA2 may serve as an additional binding receptor in these neurons. We performed the binding assay using Sema6A-Fc on WT and PlexinA4 KO DRG neurons. In the WT DRGs, a strong signal was detected along the axons, however, no signal was detected in the PlexinA4 KO neurons (*Figure 2—figure supplement 1*), indicating that PlexinA4 is the only binding and signaling receptor for Sema6A in these neurons of early E13 embryos. These in vitro experiments show that PlexinA4 mediates axonal growth cone collapse by Sema3D and Sema6A in BDNF-responsive DRG sensory neurons.

## Sema6A signaling controls sensory innervation of the mammary gland in females

The expression pattern of the Semaphorins and their in vitro activity prompted us to examine their role in mammary gland innervation. We first examined the innervation of the mammary gland in WT and Sema6A KO female and male embryos at different embryonic stages. In female embryos, Tubulin βIII staining (using Tuj1 antibody) showed more axons innervating the gland in Sema6A KO than in WT embryos, beginning at late E13 and continuing to late E14 (*Figure 3*). However, male embryos showed no difference between WT and Sema6A KO in innervation density of the gland (*Figure 3—figure supplement 1*). These results support the idea that Sema6A expressed in the epithelial cells restricts innervation of the gland by the sensory axons in females, starting at E13.

## A balance between BDNF and Semaphorin-PlexinA4 signaling controls sensory innervation of the mammary gland in females

Our findings that PlexinA4 serves as the receptor for Sema6A and Sema3D in vitro (*Figure 2*), and that mammary glands of Sema6A KO female embryos are hyperinnervated (*Figure 3*), prompted us to examine mammary gland innervation in the PlexinA4 KO embryos, as a way to look at the combined signaling from both Sema6A and Sema3D. Compared to WT female embryos, where the fiber density innervating the mammary gland increases gradually during development, in the PlexinA4 KO we detected hyperinnervation of the gland starting at E12 that remained constant at E13 and E14 (*Figure 4A,B*). Importantly, these additional axons appear to be TrkB positive (*Figure 4—figure supplement 1*). These phenotypes were much more severe than the phenotypes we detected in the Sema6A KO (*Figure 3*), suggesting that PlexinA4 indeed transmits the signaling from additional Semaphorins and has a critical role in restricting sensory innervation to the mammary gland during development. It has been shown that BDNF produced by the mammary mesenchymal cells promotes the initial ingrowth and maintenance of sensory fibers innervating the gland (*Liu et al., 2012*).

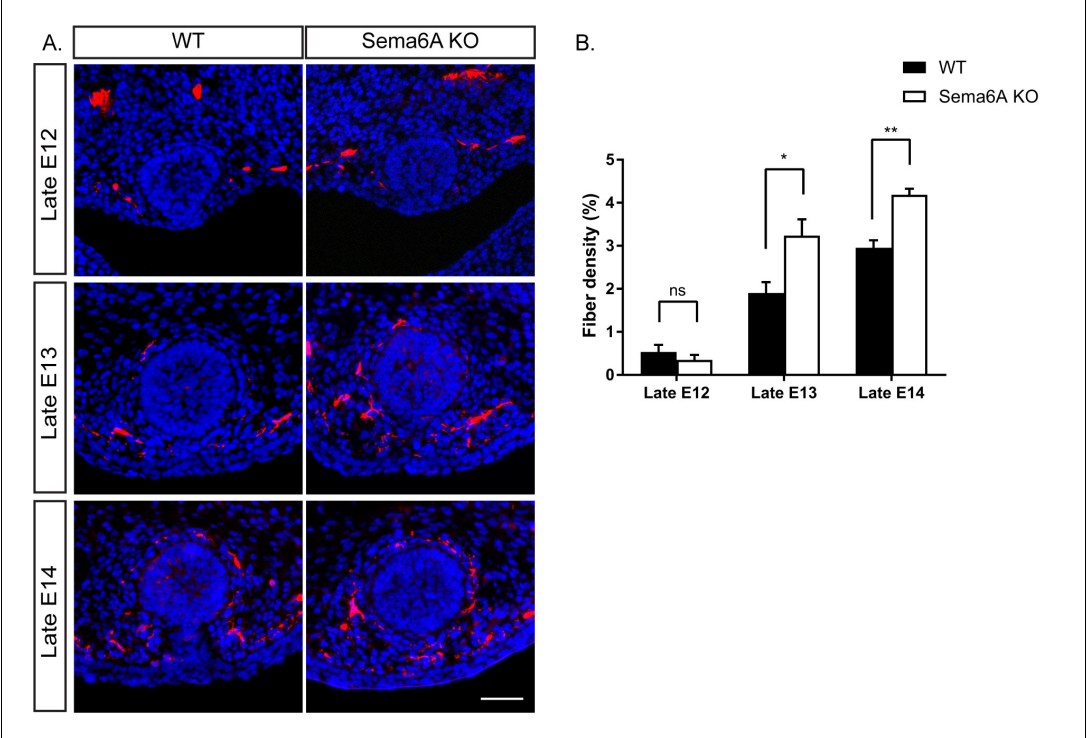

**Figure 3.** Ablation of Sema6A causes hyperinnervation of the mammary gland by sensory axons in female embryos. (**A**) Mammary gland sections of WT and Sema6A KO female embryos at different embryonic stages were stained with Tuj1 (red) and DAPI (blue) for visualization of sensory neuron innervations. Scale bar: 50 μm. (**B**) Quantification of mammary gland innervation, means ±SEM, n = 5 embryos for each bar. Late E12: p=0.3872; Late E13: p=0.019; Late E14: p=0.0026. *p<0.05, **p<0.01, ns - non-significant, two-way ANOVA followed by separate t-tests per stage.
DOI: https://doi.org/10.7554/eLife.41162.009

The following source data and figure supplements are available for figure 3:

**Source data 1.** Percentage of mammary gland innervation density in WT and Sema6A KO female embryos.
DOI: https://doi.org/10.7554/eLife.41162.012
**Figure supplement 1.** Ablation of Sema6A has no effect on mammary gland innervation in male embryos.
DOI: https://doi.org/10.7554/eLife.41162.010
**Figure supplement 1—source data 1.** Percentage of mammary gland innervation density in WT and Sema6A KO male embryos.
DOI: https://doi.org/10.7554/eLife.41162.011

Therefore, we next examined the interaction between these two pathways using a genetic approach. It has been previously demonstrated that the level of nerve growth factor (NGF) limits sensory innervation, such that NGF heterozygous (Het) animals exhibit an altered pattern of sensory projections (*Crowley et al., 1994*). Therefore, we speculated that this is the case for BDNF in the mammary gland as well. We first compared innervation of the mammary gland in female embryos heterozygous for BDNF (BDNF Het) to WT. Indeed, we observed a significant reduction in mammary gland innervation at late E12, 13 and 14 in BDNF Het embryos compared to WT (*Figure 4A,C*). Next, we crossed the BDNF Het with the PlexinA4 KO. At late E12, there was no effect on the innervation density of the gland compared to PlexinA4 KO (*Figure 4A,D*), but at later stages, reduction of BDNF levels partially suppressed the hyperinnervation. At late E14, the PlexinA4 KO/BDNF Het exhibited normal mammary gland innervation, indistinguishable from the WT embryos (*Figure 4A, D*). These results show that innervation of the mammary gland is determined by integration of Semaphorins and BDNF pathways in female embryos.

## Semaphorin-PlexinA4 signaling regulates the innervation of the mammary gland in males

In male embryos, inhibition of BDNF signaling is required for pruning of the axons innervating the mammary gland (*Liu et al., 2012*). As expected, we detected a clear developmental decrease in the

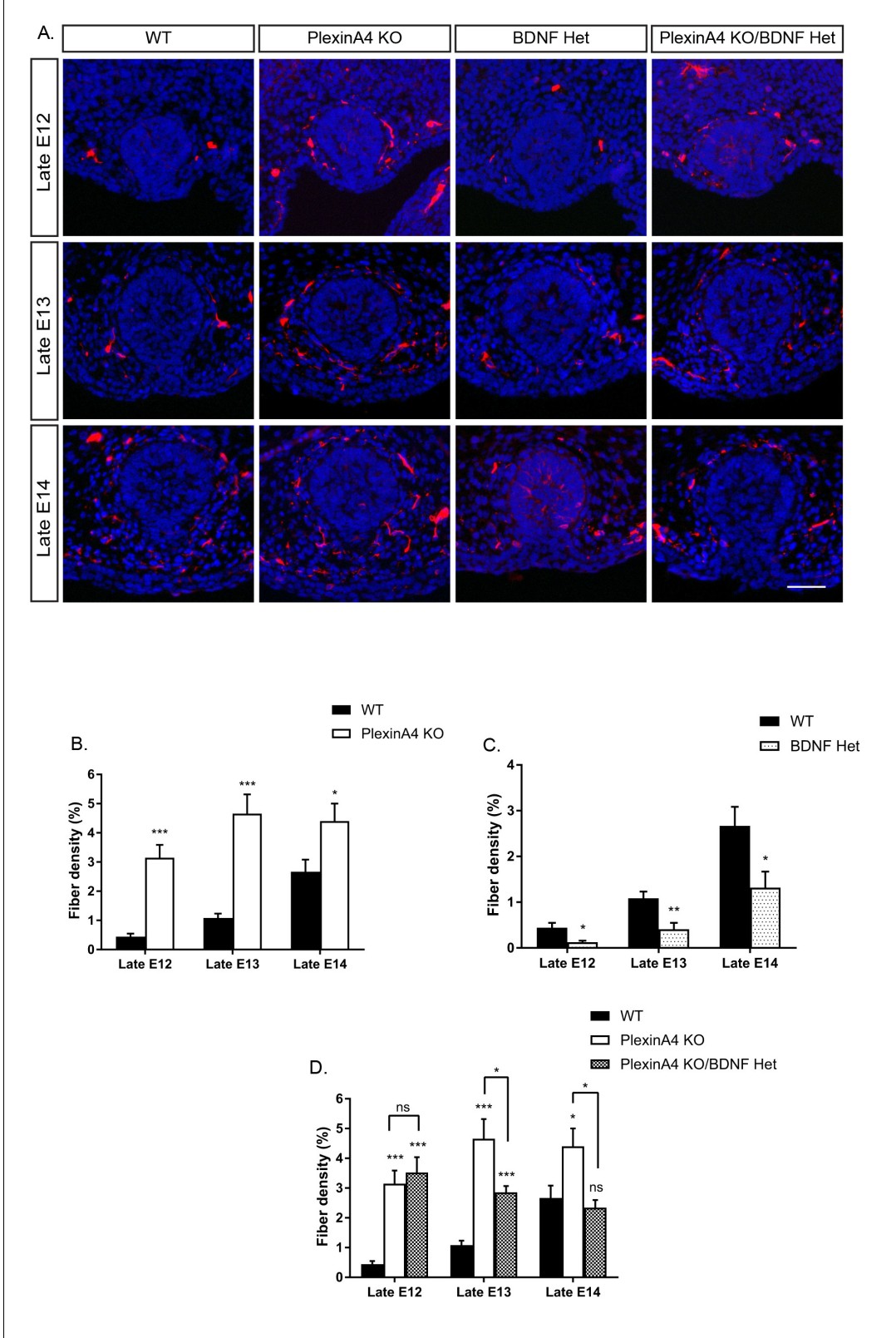

**Figure 4.** Ablation of PlexinA4 in females causes axonal hyperinnervation of the mammary gland which is balanced by genetic reduction in BDNF. (**A**) Mammary gland sections from WT, PlexinA4 KO, BDNF Het, and PlexinA4 KO/BDNF Het female embryos at different embryonic stages were stained with Tuj1 (red) and DAPI (blue) to visualize sensory neuron innervations. Scale bar: 50 μm. (**B–D**) Quantification of mammary gland innervation, means ± SEM. Late E12: n = 9, 7, 8, 11; Late E13: n = 9, 10, 6, 8; Late E14: n = 5, 8, 7, 6 for WT, PlexinA4 KO, BDNF Het and PlexinA4 KO/BDNF

*Figure 4 continued on next page*

Figure 4 continued

Het, respectively. (B) WT vs PlexinA4 KO: Late E12 p<0.0001, Late E13 p<0.0001, Late E14 p=0.0484. (C) WT vs BDNF Het: Late E12 p=0.0178, Late E13 p=0.0084, Late E14 p=0.0359. (D) WT vs PlexinA4 KO/BDNF Het: Late E12 p<0.0001, Late E13 p=0.0003, Late E14 p=0.8856. PlexinA4 KO vs PlexinA4 KO/BDNF Het: Late E12 p=0.9940, Late E13 p=0.0480, Late E14 p=0.0149. *p<0.05, **p<0.01, ***p<0.001, ns – non-significant, two-way ANOVA followed by separate t-tests per stage. P values compared to WT are marked on top of each bar.
DOI: https://doi.org/10.7554/eLife.41162.013

The following source data and figure supplement are available for figure 4:

Source data 1. Percentage of mammary gland innervation density in WT, PlexinA4 KO, BDNF Het and PlexinA4 KO/BDNF Het female embryos.
DOI: https://doi.org/10.7554/eLife.41162.015

Figure supplement 1. Excess innervations in female PlexinA4 KO are TrkB positive.
DOI: https://doi.org/10.7554/eLife.41162.014

innervation of WT male mammary glands between late E12 and late E14 (*Figure 5A,B*). To test whether Semaphorin-PlexinA4 signaling has any role in regulating developmental axonal pruning in males, we examined the innervation of the mammary gland in the PlexinA4 KO male embryos. Similar to females, we detected hyperinnervation at late E12 (*Figure 5B*), suggesting that PlexinA4 signaling restricts the initial innervation of the mammary gland in males as well. Interestingly, there was virtually no decrease in fiber density between late E12 and late E13 in the PlexinA4 KO. In comparison, in WT males, fiber density decreased by about 50% during the same time frame (*Figure 5B*). Similar to our results from female embryos, these axons are TrkB positive (*Figure 5—figure supplement 1*). Between late E13 and late E14 there was a sharp decrease in fiber density in the PlexinA4 KO, but to a level which is still significantly higher than that of WT embryos. Importantly, the delay in pruning in the PlexinA4 KO embryos was not due to lower expression of TrkB.T1, as we detected similar expression levels in the mesenchymal cells of WT and PlexinA4 KO embryos, using the TrkB$^{ECD}$ antibody (*Figure 5—figure supplement 2*). Since the initial innervation in the PlexinA4 KO at late E12, prior to pruning onset, is much higher than in the WT, we could not discriminate between two possibilities: 1) the delay in pruning is due to increased innervation at the beginning of the process or 2) that signaling through PlexinA4 is required for pruning. To address this issue, we compared the innervation of the male mammary gland on the background of BDNF Het. As in the females, there is a clear reduction in mammary gland innervation in the BDNF Het male embryos compared to WT at late E12 (*Figure 5A,C*). Unlike in females, we detected a reduction in innervation (~50%) in the PlexinA4 KO/BDNF Het male embryos compared to the PlexinA4 KO, as early as late E12. Despite this greatly reduced initial innervation, there was still a strong delay in axonal elimination between late E12 and late E13 (only 20% decrease). By late E14, we detected similar levels of innervation in the PlexinA4 KO/BDNF Het and PlexinA4 KO embryos, which was still significantly higher than the WT (*Figure 5A,D*). Overall, these results indicate that PlexinA4 is required for the normal progression of developmental axonal pruning of fibers innervating the male mammary gland.

## Decreased BDNF levels promote Semaphorin-PlexinA4-induced growth cone collapse in vitro

Our in vivo analysis suggests that balance of BDNF and Semaphorin-PlexinA4 signaling controls the innervation of the mammary gland. To further study this balance, we carried out in vitro growth cone collapse experiments under conditions of reduced BDNF levels. DRG explants of early E13 embryos were grown in the presence of BDNF for 48 hr, then the media was replaced to media containing BDNF (+BDNF), or media without BDNF (-BDNF), both with low concentrations of Sema6A (0.025 ng/μl) for 30 min. In the presence of BDNF, this concentration of Sema6A did not induce growth cone collapse. However, adding Sema6A at this concentration to the -BDNF media resulted in 60% growth cone collapse (*Figure 6*). This demonstrates that at low levels of BDNF, sensory axons become hypersensitive to PlexinA4 signaling, which results in more collapsed growth cones. These results support the idea that BDNF inhibits the Semaphorin signaling, and may suggest that upon BDNF sequestering the axons become hypersensitive to PlexinA4 signaling.

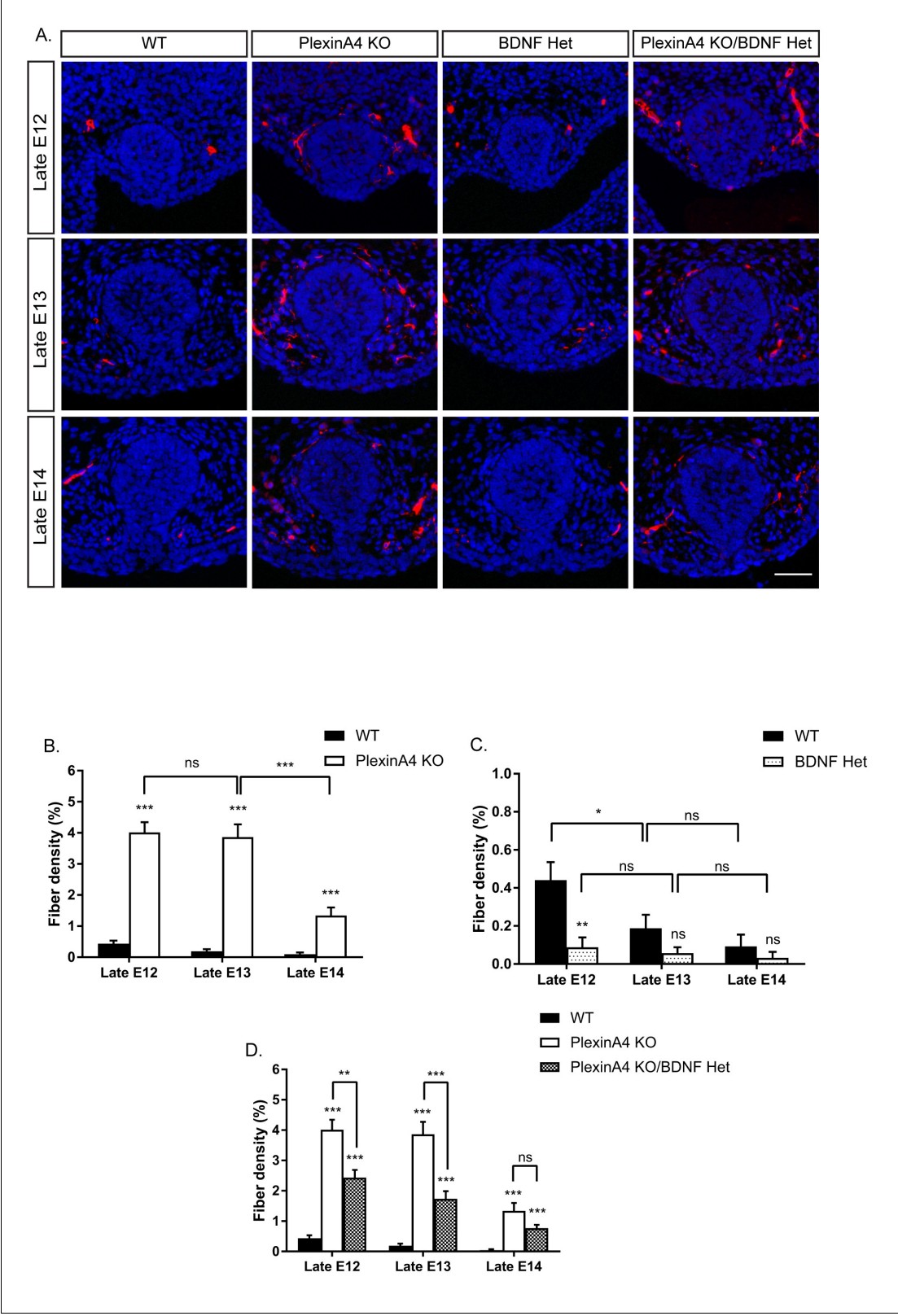

**Figure 5.** Ablation of PlexinA4 delays axonal pruning in males independent of the initial levels of the mammary gland innervation. (**A**) Mammary gland sections of WT, PlexinA4 KO, BDNF Het, and PlexinA4 KO/BDNF Het male embryos at different embryonic stages were stained with Tuj1 (red) and DAPI (blue) to visualize sensory neuron innervations. Scale bar: 50 μm. (**B–D**) Quantification of mammary gland innervation, means ±SEM. Late E12: n = 11, 9, 9, 10; Late E13: n = 10, 7, 7, 8; Late E14: n = 6, 11, 6, 6 for WT, PlexinA4 KO, BDNF Het and PlexinA4 KO/BDNF Het, respectively. (**B**) WT vs

*Figure 5 continued on next page*

Figure 5 continued

PlexinA4 KO: Late E12 p<0.0001, Late E13 p<0.0001, Late E14 p<0.0001. PlexinA4 KO: Late E12 vs Late E13 p=0.7803, Late E13 vs Late E14 p<0.0001.
(C) WT vs BDNF Het: Late E12 p=0.0013, Late E13 p=0.1156, Late E14 p=0.3739. WT: Late E12 vs late E13 p=0.0473, Late E13 vs late E14 p=0.3282.
BDNF Het: Late E12 vs Late E13 p=0.6117, Late E13 vs Late E14 p=0.5833 (D) WT vs PlexinA4 KO/BDNF Het: Late E12 p<0.0001, Late E13 p<0.0001,
Late E14 p=0.00011. PlexinA4 KO vs PlexinA4 KO/BDNF Het: Late E12 p=0.0010, Late E13 p=0.0003, Late E14 p=0.1049. *p<0.05, **p<0.01,
***p<0.001, ns – non-significant, two-way ANOVA followed by separate t-tests per stage. P values compared to WT are marked on top of each bar.
DOI: https://doi.org/10.7554/eLife.41162.016

The following source data and figure supplements are available for figure 5:

**Source data 1.** Percentage of mammary gland innervation density in WT, PlexinA4 KO, BDNF Het and PlexinA4 KO/BDNF Het male embryos.
DOI: https://doi.org/10.7554/eLife.41162.019
**Figure supplement 1.** Excess innervations in male PlexinA4 KO are TrkB positive.
DOI: https://doi.org/10.7554/eLife.41162.017
**Figure supplement 2.** PlexinA4 does not control the expression of TrkB.T1.
DOI: https://doi.org/10.7554/eLife.41162.018

## PlexinA4 and TrkB.T1 operate in parallel and converge to an axonal elimination pathway

Since PlexinA4 does not regulate the expression of TrkB.T1, which neutralizes BDNF in the mammary gland of male embryos (*Figure 5—figure supplement 2*), we tested if PlexinA4 and TrkB.T1 operate in parallel pathways. To assess interaction between these two pathways, we used a genetic approach in which we analyzed sensory innervation of the mammary gland in PlexinA4/TrkB.T1 double KO in late E13 and E14 embryos (*Figure 7*). In the single PlexinA4 KO embryos at late E13 and late E14, mammary innervation is significantly greater than in the WT, but between E13 and E14, innervation of the gland decreases to levels that are nonetheless still elevated compared to the WT. In the TrkB.T1 single KO at E13, the level of innervation is higher than the WT due to decreased pruning, as previously demonstrated by Liu et al. However, at E14, the level of innervation dropped to WT levels. There was no significant difference in innervation between the double KO and the PlexinA4 single KO at E13 or E14. Overall, these results are consistent with the idea that TrkB.T1 and PlexinA4 operate in parallel and converge to the same axonal elimination pathway.

## Discussion

The process of target innervation by sensory axons during embryonic development is largely regulated by neurotrophic factors secreted by the target in limited amounts. Studies of NGF have demonstrated a strong correlation between its expression and the innervation density of the skin (*Davies et al., 1987*), and that overexpression of NGF results in hyperinnervation of the target organ (*Albers et al., 1994*). However, there is growing evidence that additional target-derived cues are involved in specifying innervation, some through interaction with the neurotrophin signaling pathways (*Guha et al., 2004*; *Bäumer et al., 2014*; *Wheeler et al., 2014*).

The mammary gland serves as a unique model for sexually dimorphic target innervation controlled by BDNF, but whether other factors regulate this process jointly with BDNF was not known. Through the studies described here, we discovered a critical role for the Semaphorin-PlexinA4 signaling pathway in axonal innervation restriction and possibly pruning promotion.

Our studies uncovered a non-dimorphic expression of several members of the Semaphorin family in the mammary gland that signal through PlexinA4. Since the Androgen receptor is not expressed in DRG neurons (*Liu et al., 2012*), it is likely that PlexinA4 is expressed in non-dimorphic manner as well.

In vivo ablation of PlexinA4 caused hyperinnervation of the gland in female embryos. This effect of PlexinA4 ablation was reversed by genetic reduction of BDNF levels in PlexinA4 KO/BDNF Het embryos, which exhibited normal innervation of the mammary gland, indistinguishable from WT levels, at late E14. These results suggest that although BDNF is expressed in limited amounts, its physiological levels are sufficient to sustain high levels of innervation, which are curbed by Semaphorin signaling.

In male embryos, ablation of PlexinA4 resulted in initial hyperinnervation of the gland. Co-reduction of BDNF with PlexinA4 ablation dramatically reduced the initial hyperinnervation at late E12,

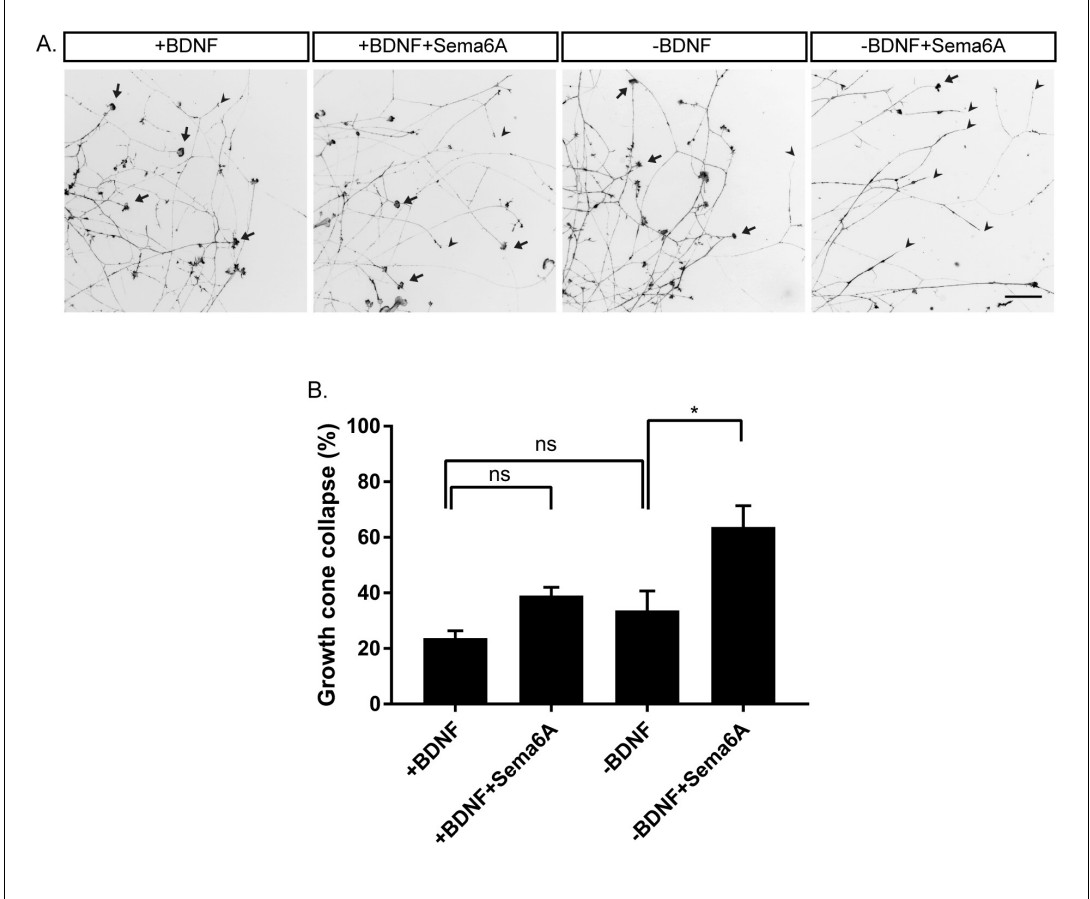

**Figure 6.** Hypersensitivity to PlexinA4 signaling induced by reduction in BDNF. (**A**) Explants of early E13 embryonic DRGs were cultured in media containing 50 ng/ml BDNF for 48 hr, then the media was replaced to 50 ng/ml BDNF (+BDNF), or media without BDNF (-BDNF) both with low concentration of Sema6A (0.025 ng/μl) for 30 min. Growth cones were visualized with Phalloidin- Rhodamine staining. Black arrowheads refer to collapsed growth cones and arrows indicate non-collapsed. Scale bar: 50 μm. (**B**) Quantification of percent of collapsed growth cones, means ± SEM of more than 1000 growth cones obtained from three independent experiments. +BDNF vs +BDNF+Sema6A p=0.2872, +BDNF vs -BDNF p=0.6095, -BDNF vs -BDNF+Sema6A p=0.0223. *p<0.05, ns – non-significant, one-way ANOVA with *post hoc* analysis.
DOI: https://doi.org/10.7554/eLife.41162.020

The following source data is available for figure 6:

**Source data 1.** Percentage of collapsed growth cones.
DOI: https://doi.org/10.7554/eLife.41162.021

although this was not observed in females at the same stage. The basis for this difference is currently unclear, however, it may reflect an intrinsic hypersensitivity of the male axons to BDNF levels, which is only revealed on the background of PlexinA4 KO. This hypersensitivity may help to promote pruning once BDNF is sequestered. We detected delayed pruning of the fibers innervating the gland in the PlexinA4 KO and the PlexinA4 KO/BDNF het, supporting a model in which PlexinA4 promotes pruning independently of the initial levels of innervation. However, we cannot exclude the idea that the excess initial axonal innervation by itself generates a protective mechanism against pruning, for example by axon-axon adhesion as was observed in flies (*Bornstein et al., 2015*). Lastly, while our results show that ablation of PlexinA4 does not affect the expression of TrkB.T1, we were not able to visualize BDNF at the gland by immunostaining. Therefore, we cannot completely dismiss the possibility that ablation of PlexinA4 affects the expression of BDNF.

Using in vitro growth cone collapse assays with low levels of BDNF resulted in hypersensitivity to PlexinA4 signaling, showing that inhibition of BDNF signaling renders the axons more sensitive to PlexinA4. Although this assay is mainly correlated with growth inhibition by the Semaphorins and not axonal pruning, a recent study suggested that growth cone collapse is an early and critical step

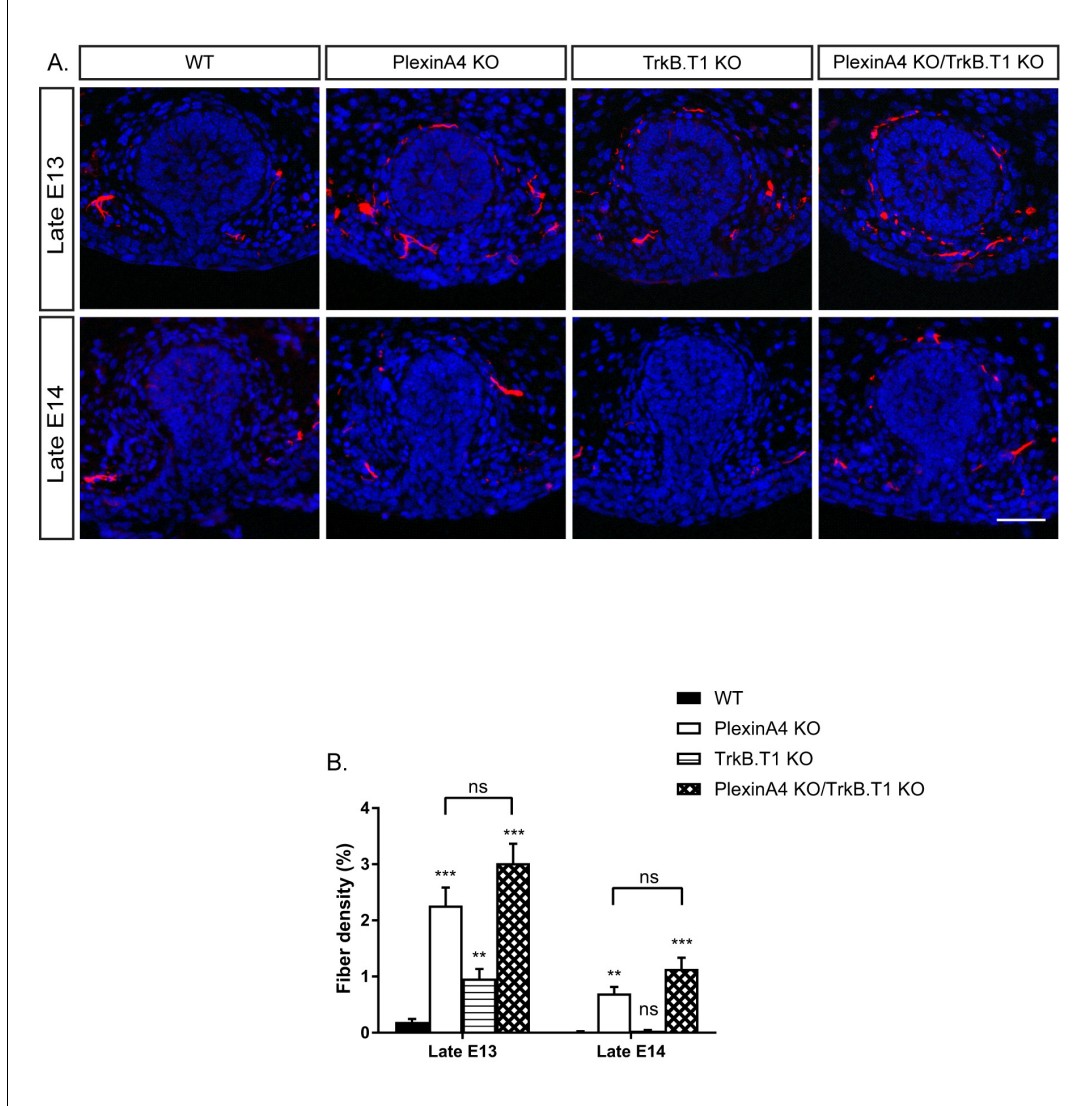

**Figure 7.** Co-ablation of TrkB.T1 and PlexinA4 has no additive effect on axonal pruning. (**A**) Mammary gland sections of WT, PlexinA4 KO, TrkB.T1 KO, and PlexinA4 KO/TrkB.T1 KO male embryos at different embryonic stages were stained with Tuj1 (red) and DAPI (blue) to visualize sensory neuron innervations. Scale bar: 50 μm. (**B**) Quantification of mammary gland innervation, means ±SEM. Late E13: n = 5, 8, 6, 7, Late E14: n = 5, 5, 9, 6 for WT, PlexinA4 KO, TrkB.T1 KO and PlexinA4 KO/TrkB.T1 KO, respectively. WT vs PlexinA4 KO: Late E13 p<0.0001, Late E14 p=0.0015. WT vs TrkB.T1 KO: Late E13 p=0.0020, Late E14 p=0.0867. WT vs PlexinA4 KO/TrkB.T1 KO: Late E13 p<0.0001, Late E14 p=0.0006. PlexinA4 KO vs PlexinA4 KO/TrkB.T1 KO: Late E13 p=0.1114, Late E14 p=0.1037. **p<0.01, ***p<0.001, ns – non-significant, two-way ANOVA followed by separate t-tests per stage. P values compared to WT are marked on top of each bar.

DOI: https://doi.org/10.7554/eLife.41162.022

The following source data is available for figure 7:

**Source data 1.** Percentage of mammary gland innervation density in WT, PlexinA4 KO, TrkB.T1 KO and PlexinA4 KO/TrkB.T1 KO male embryos.
DOI: https://doi.org/10.7554/eLife.41162.023

in axonal degeneration in response to trophic deprivation (*Unsain et al., 2018*). Therefore, the hypersensitivity to PlexinA4-induced growth cone collapse may reflect promotion of the pruning process.

Interestingly, while BDNF counteracts PlexinA4 effects, our results suggest that PlexinA4 and the BDNF receptor isoform TrkB.T1 operate in the same pathway of axonal elimination, since co-ablation of the two did not have any additive effect. This result supports previous data that TrkB.T1 antagonizes BDNF signaling in the mammary gland (*Liu et al., 2012*). Nevertheless, neither ablation of

PlexinA4 or of TrkB.T1 was able to prevent pruning at late E14, when the innervation density drops sharply. During mammary organogenesis, the mammary gland of male embryos starts to regress as a result of Androgen release by the male gonads at E13 (*Robinson, 2007*), and this regression probably forces elimination of the axons.

Altogether, our study shows that developmental innervation of the mammary gland is regulated by a balance between BDNF, which promotes the initial ingrowth of the sensory fibers, and constant Semaphorin signaling, which restricts innervation (*Figure 8A*). Upon inhibition of BDNF-TrkB signaling, Semaphorin-PlexinA4 signaling may promote axonal pruning (*Figure 8B*).

A previous in vitro study has demonstrated that the class 3 Semaphorins inhibit neurotrophin signaling and that this inhibition is a part of the mechanism by which they induce growth cone collapse (*Atwal et al., 2003*). Our results also support the idea that neurotrophins signaling suppresses the Semaphorin signaling, and once this suppression is relieved, the axons become hypersensitive to Semaphorins, as was proposed by Dontchev and Letourneau (*Dontchev and Letourneau, 2002*). Overall, our work provides the first clear evidence that the balance between these two cues plays a

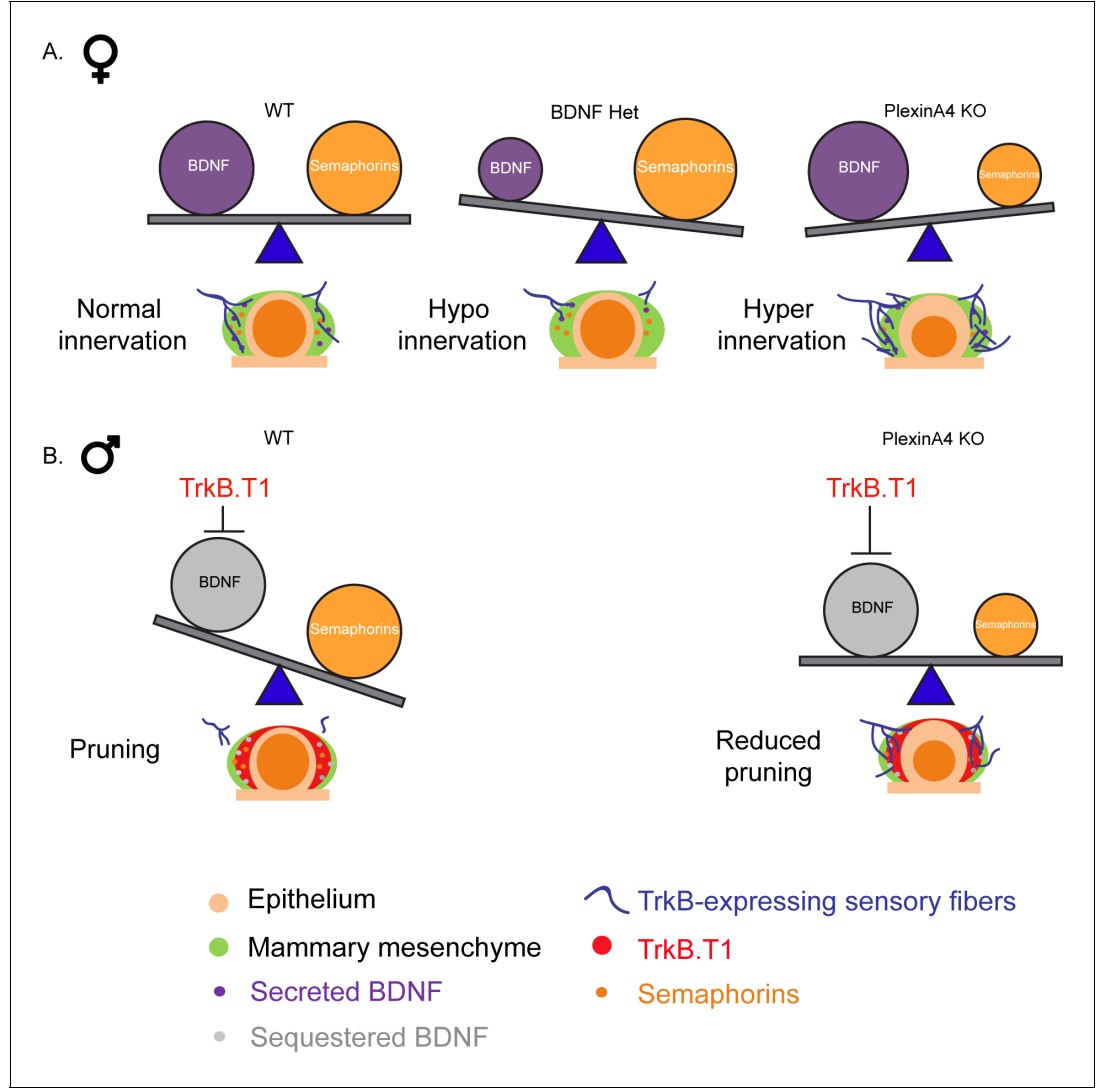

**Figure 8.** The relative balance between BDNF and Semaphorins regulates the level of mammary gland innervation in females and may promote axonal pruning in males. (**A**) In females: a balance between BDNF and Semaphorins specifies normal innervation of the mammary gland. Decreased levels of BDNF cause hypoinnervation and decreased Semaphorins cause hyperinnervation. (**B**) In males: Sequestering of BDNF by TrkB.T1 results in pruning, while decreased Semaphorin signaling results in reduced pruning.
DOI: https://doi.org/10.7554/eLife.41162.024

role in accurate wiring of the nervous system during development, and that physiologically tipping this balance may induce axonal pruning.

The Semaphorins have been previously implicated in pruning of hippocampal axons, where developmental expression of Sema3F was proposed to largely regulate this process (*Bagri et al., 2003*). In contrast, in the mammary gland, we did not detect any dimorphic or temporal expression of Semaphorins or PlexinA4; instead, our results suggest that their ability to prune axons in male embryos is gated by inhibition of BDNF signaling. Whether there are equivalent mechanisms that gate Semaphorin-dependent axonal pruning in the hippocampus remains to be discovered.

Expression of Semaphorins and Plexins has been previously demonstrated in the developing mammary gland postnatally, and was suggested to affect ductal growth and morphogenesis of the gland (*Morris et al., 2006*). Here we present new data that show expression of Sema3D and Sema6A in the mammary epithelium during embryonic development, and their involvement in regulation of the mammary gland innervation through the PlexinA4 receptor. We also detected expression of Sema3F in the mammary epithelium, but in vitro it had no effect on growth cone collapse of BDNF-responsive sensory neurons, suggesting it is not involved in target innervation. It may have a role in the regulation of ductal directional outgrowth and branching morphogenesis during later development. This is supported by the fact that Neuropilin2, which serves as the binding receptor of Sema3F (*Yoshida, 2012*), promotes postnatal branching morphogenesis of the mammary gland (*Goel et al., 2011*). Together, these results suggest that the Semaphorins play a dual role in mammary gland development, determining the extent of sensory innervation density prenatally and promoting gland morphogenesis postnatally.

Intriguingly, the neurotrophins and the Semaphorins continue to be expressed in peripheral tissues throughout the life of the organism, and their expression may be altered in pathological conditions (*Apfel, 1999*; *Yamaguchi et al., 2008*; *Tominaga et al., 2009*). Our findings raise the possibility that altered expression of these cues could induce aberrant innervation in mature animals, causing either hypersensitivity or reduced sensation, depending on the balance between these pathways.

# Materials and methods

## Key resources table

| Reagent type (species) or resource | Designation | Source or reference | Identifiers | Additional information |
|---|---|---|---|---|
| Genetic reagent (*M. Musculus*) | Sema6a$^{-/-}$ | PMID: 11242070 | | Dr. Marc Tessier-Lavigne (Stanford) |
| Genetic reagent (*M. Musculus*) | Plxna4$^{-/-}$ | PMID: 15721238 | | Dr. Marc Tessier-Lavigne (Stanford) |
| Genetic reagent (*M. Musculus*) | Ntrk2.T1$^{-/-}$ | PMID: 16815329 | | Dr. Lino Tessarollo (NIH) |
| Genetic reagent (*M. Musculus*) | Bdnf$^{+/-}$ | Jackson Laboratory | stock #: 002266 RRID: MGI:2175720 | PMID: 8139657 |
| Cell line (Homo sapiens) | HEK293 | ATCC | ATCC CRL-1573 RRID:CVCL_0045 | |
| Cell line (Cercopithecus aethiops) | COS1 | ATCC | ATCC CRL-1650 RRID: CVCL_0223 | |
| Antibody | Mouse monoclonal anti-tubulin β-III | R and D | Cat. #: MAB1195 RRID:AB_357520 | IHC (1:500) |
| Antibody | Rabbit polyclonal anti-tubulin β-III | Biolegend | Cat. # 802001 RRID:AB_2564645 | IHC-Fr (1:1000) |
| Antibody | Polyclonal Goat anti-Mouse Trkb | R and D | Cat. # AF1494 RRID:AB_2155264 | IHC-Fr (1:100) |
| Peptide, recombinant protein | Human Sema6A-FC | PMID: 20606624 | | |
| Peptide, recombinant protein | Mouse Sema3D-FC | This paper | | |

*Continued on next page*

*Continued*

| Reagent type (species) or resource | Designation | Source or reference | Identifiers | Additional information |
|---|---|---|---|---|
| Peptide, recombinant protein | Mouse AP-Sema3F | PMID: 9331348 | | |
| Chemical compound, drug | Phalloidin-Rhodamine | Sigma | p1951 | 1:250 |
| Chemical compound, drug | Ara-C | Sigma | C1768 | 30 uM |

## Mouse lines

Pregnant female ICR mice (8–12 week old) were purchased from Envigo (Rehovot, Israel). The Sema6A mutant (*Leighton et al., 2001*) and PlexinA4 KO (*Yaron et al., 2005*) lines have been described previously. TrkB.T1 KO mice were kindly provided by the lab of Lino Tessarollo and generated as described previously (*Dorsey et al., 2006*). Mice heterozygous for the Bdnf$^{tm1Jae}$ mutation (JAX stock #002266) (*Ernfors et al., 1994*) were purchased from The Jackson Laboratory. All mice were of mixed strains, and colonies were bred and maintained at the Department of Veterinary Resources of the Weizmann Institute of Science, Israel, according to recommendations of the Federation of European Laboratory Animal Science Associations.

## Embryo stages

Staged embryos were generated by interbreeding mice of the appropriate genotype in overnight mating. The day on which vaginal plug was observed was counted as embryonic day 0. Pregnant females were sacrificed to harvest the embryos at different times of a given embryonic day: E12, E13 or E14. Early of the day was done at 9–10 and late at 16–17 as was previously described (*Liu et al., 2012*). Embryos were genotyped by PCR using tail DNA samples, and the sex was determined by using primers of the Zfy1 gene, as was previously described (*Liu et al., 2012*).

## Immunohistochemistry- Paraffin and frozen sections

For sensory neurons staining: embryos at different developmental stages were fixed in 4% formaldehyde overnight (O.N) at room temperature, followed by paraffin embedding. Transversal sections of 4 μm from the first pair of mammary glands were taken for immunostaining. The sections underwent deparaffinization with xylene and ethanol, followed by antigen retrieval using 10 mM sodium citrate buffer pH=6 at 125°C in a decloaking chamber. Sections were blocked with 1% bovine serum albumin (BSA) and 0.5% goat serum in PBST (0.1% Triton X-100 in PBS) for 1 hr at room temperature, followed by O.N incubation with mouse anti-tubulin β-III antibody (R&D, Tuj1 clone, MAB1195, 1:500) in 1% BSA. At least five embryos were analyzed for each condition.

For TrkB$^{ECD}$ staining: embryos were fixed in 4% formaldehyde O.N at 4°C, followed by cryoprotection with 30% sucrose in PBS at 4°C for O.N incubation. Embryos were embedded in O.C.T (Tissue Tek), frozen at -80°C, and sectioned at 20 μm. Transversal sections of the first pair of mammary glands were dried O.N at room temperature, washed with PBS and blocked with 5% donkey serum in PBST for 1 hr, followed by incubation with goat anti- TrkB$^{ECD}$ (R&D, AF1494, 1:100), diluted in blocking solution overnight at 4°C. The next day, sections were washed with PBS and incubated with secondary antibodies (Jackson Immunoresearch, 1:200) and DAPI diluted in PBS for 1–2 hr, washed again with PBS, and mounted with fluoromount-G (Southern Biotech). At least two embryos were analyzed for each sex.

Pictures of Tuj1 staining were taken using 90i upright fluorescent microscope (Nikon). Pictures of TrkB$^{ECD}$ were taken with 60X oil (UIS2) objective using IX81-based Olympus FluoView 1000 laser confocal scanning microscope.

## Quantification of the percentage of fiber density innervating the mammary gland

Images of transversal paraffin sections of the first pair of mammary glands stained with anti-Tuj1 and DAPI were processed and analyzed using ImageJ software. The measurement of fiber density was done as previously described (*Liu et al., 2012*). In short a 15 μm band was drawn surrounding the

mammary epithelium according to the DAPI staining, and the percent of Tuj1 positive pixels within this band was measured using the area fraction measurement, giving the percent of fiber density.

Overall in this study 208 embryos were analyzed. Outliers were considered as data-points that are more than four standard deviations from the mean and were excluded from the data. Differences between groups and stages were tested with a 2-way ANOVA on log-transformed measurements. In cases where the interaction between group and stage was significant, we followed the ANOVA with separate t-tests per stage, to elucidate in which stage the differences between groups was significant.

### In situ hybridization

Fluorescent ISH on transversal paraffin sections of the first pair of mammary glands from ICR female and male embryos at late E13 was performed as previously described (*Shwartz and Zelzer, 2014*). The pattern of expression of different genes of the Semaphorins was detected using a digoxigenin (DIG)-labeled probes. Two embryos were analyzed for each probe with similar results.

### Cell culture and transfection

HEK293 and COS1 cells were originally obtained from the ATCC –The Global Bioresource Center (no authentication was performed). Cell lines were cultured in DMEM (Gibco) supplemented with 10% FBS, 100 U/mL penicillin-streptomycin, 2 mM glutamine and Biomyc-3 anti-mycoplasma solution, at 37°C, 95% humidity and 5% CO2. The cells were not tested for Mycoplasma. Cells were transfected at 70% confluency in 10 cm plates using Polyjet transfection reagent with 5 µg of the different plasmids.

### Production, purification and quantification of recombinant Semaphorin fusion proteins

#### Sema6A-Fc

COS1 cells were transfected with a plasmid containing the extra-cellular region of Sema6a fused to the human Fc, as was previously described (*Haklai-Topper et al., 2010*), using standard protocol. Twenty-four hours after transfection, medium was replaced with serum-free Dulbecco's Modified Eagle Medium/Ham's F12 (DMEM-F12), which was collected after an additional 48 hr. Conditioned media went through HPLC purification using protein A column (HiTrapTM rProtein A FF, GE), and concentration of the Semaphorin was determined using SDS-PAGE followed by Coomassie staining compared to known concentrations of BSA.

#### Sema3D-Fc

Human embryonic kidney (HEK293T) cells were transfected with a plasmid containing the Mouse Sema3D coding sequence fused to the human Fc, using standard protocol. Twenty-four hours after transfection, medium was replaced with serum-free DMEM-F12, which was collected after an additional 48 hr. Conditioned media went through HPLC purification using protein A column (HiTrap rProtein A FF, GE), and concentration of the Semaphorin was determined using SDS-PAGE followed by Coomassie staining compared to known concentrations of BSA.

#### AP-Sema3F

COS1 cells were transfected with an AP (Alkaline Phosphatase)-Sema3F expression vector using standard protocol. Twenty-four hours after transfection, medium was replaced with serum-free DMEM-F12, which was collected after an additional 48 hr. The conditioned medium was concentrated 30 times using Vivaspin 20 (Sartorius, 30,000 MWCO), aliquoted and stored at -80°C. Concentration of AP-Sema3F in the conditioned media was quantified by supplementation of the concentrated conditioned media with AP's chromogenic substrate *para*-Nitrophenylphosphate (Sigma) followed by spectrophotometry measurement of optical density at a wavelength of 405 nm using Ultrospec 2100 Pro (Amersham Biosciences).

### Growth cone collapse assay

DRGs from early E13 embryos of the appropriate genotype were dissected and plated as explants in chambers coated with 10 µg/ml Poly-D-lysine (PDL) and 10 µg/ml Laminin, and contained

Neurobasal-A growth medium supplemented with 2% B-27, 1% Penicillin-Streptomycin, 1% L-Gluta-mine (NB), and 50 ng/ml BDNF (Peprotech). Twenty-four hours post plating, the medium was replaced to a one containing 30 μM Cytosine β-D-arabinofuranoside (Ara-C) and after additional 24 hr treated with 1 ng/μl of Sema6A-Fc/0.02 ng/μl of Sema3D-Fc/1 nM of AP-Sema3F for 30mins. The control for the HPLC-purified Semaphorins (Sema6A-Fc and Sema3D-Fc) was NB with BDNF, and for AP-Sema3F was conditioned medium of AP empty vector transfected cells concentrated the same way. For collapse assay without BDNF, 24 hr after adding the Ara-C the medium was replaced to NB or NB +50 ng/ml BDNF with or without 0.025 ng/μl of Sema6A-Fc for 30mins.The cultures were fixed with 4% Paraformaldehyde +30% Sucrose solution, and stained with Phalloidin-Rhodamine (Sigma-Aldrich, 1:250) for visualization of F-actin filopodia. Growth cones with no or few filopodia were considered as collapsed. Percentage of collapsed growth cones was calculated for each treatment group and control. A minimum of three independent experiments were performed per condition, in each experiment three DRGs were analyzed per embryo. Percentages are means between experiments. Error bars represent the SEM between replicates. Statistical analysis was performed using one-way or two-way ANOVA followed by *post hoc* analysis (Prism7 software).

## Binding assay of Sema6A-Fc

DRGs from early E13 embryos of the appropriate genotype were dissected and plated as explants in chambers coated with 10 μg/ml Poly-D-lysine (PDL) and 10 μg/ml Laminin, and contained Neuroba-sal-A growth medium supplemented with 2% B-27, 1% Penicillin-Streptomycin, 1% L-Glutamine, and 50 ng/ml BDNF (Peprotech). 48 hr post plating, the DRGs were washed with binding buffer (Hank's-Buffered Salt Solution with 0.2% BSA, 0.1% $NaN_3$, 5 mM $CaCl_2$, 1 mM $MgCl_2$ and 20 mM HEPES, pH=7.0) for 10 min and incubated for 90 min at room temperature with 1 ng/μl of Sema6A-Fc containing 1:1000 goat anti-human IgG (Fc specific)- Alkaline Phosphates (AP)-conjugated antibody (A9544, Sigma) in binding buffer. After the removal of unbound ligand, DRGs were fixed with 4% PFA +30% Sucrose for 12 min and rinsed three times with 20 mM HEPES pH=7.0 and 150 mM NaCl. In order to destroy intrinsic AP activity, DRGs were heat inactivated in 65°C water bath for 30 min. Finally, DRGs were rinsed three times with AP buffer (100 mM TRIS PH=9.5, 100 mM NaCl and 50 mM $MgCl_2$) and the AP activity was evaluated by the formation of precipitants after an overnight incubation with 1:50 Nitro blue tetrazolium chloride/5-bromo-4-chloro-3-indolyl phosphate substrate (NBT/BCIP, Roche) in AP buffer.

## X-gal staining

For whole-mount staining of β-galactosidase (β-gal) activity, embryos at different developmental stages were fixed in 4% formaldehyde for 2 hr in 4°C, followed by three washes in PBS. Staining was performed with X-gal solution (5 mM Potassium ferricyanide, 5 mM Potassium ferrocyanide, 2 mM $MgCl_2$ and 0.5 mg/ml 5-bromo-4-chloro-3-indolylβ-D-galactopyranoside [X-gal] in PBS). Incubation time was 2–4 hr in room temperature, depending on the developmental stage, followed by three washes in PBS. The reaction was stopped by O.N incubation with 4% formaldehyde solution in 4°C.

## Antibodies and reagents

For immunohistochemical stainings the following primary antibodies were used: Goat anti-TrkB[ECD] (R&D, AF1494, 1:100); Mouse anti-tubulin βIII (R&D, TUJ-1 clone/catalog, 1:500). Fluorescent secondary antibodies were purchased from Jackson Immunoresearch: donkey anti-goat Cy3 (1:200), and goat anti-mouse Cy3 (1:200). Phalloidin-Rhodamine was used at 1 ug/ml (Sigma-Aldrich, P1951).

## Acknowledgements

We thank the Yaron lab members for advice and criticism, Lino Tessarollo (NIH) for providing us with the TrkB.T1 KO line, Eran Horenstein's lab for the use of the microtome, Eli Zelzer's lab for the help with all the in-situ experiments, Vladimir Kiss for the help with the confocal microscope, Ron Rotkopf for excellent statistical assistance and Oren Schuldiner for critically reading the manuscript. This work was supported by funding to AY from the Israel Science Foundation (873/14) and the Canadian Institutes of Health Research (CIHR), the International Development Research Centre (IDRC), the Israel Science Foundation (ISF) and the Azrieli Foundation (2412/15), The Nella and Leon Benoziyo Center for Neurological Diseases, The Y Leon Benoziyo Institute for Molecular Medicine, Adelis

Foundation, Mont-Royal Trust, The Irving B Harris Fund for New Directions in Brain Research, the Joseph D Shane Fund for Neurosciences, and Mr. and Mrs. James Orleans. AY is an incumbent of the Jack and Simon Djanogly Professorial Chair in Biochemistry.

## Additional information

### Funding

| Funder | Grant reference number | Author |
| --- | --- | --- |
| Israel Science Foundation | 873/14 | Avraham Yaron |
| Canadian Institutes of Health Research | 2412/15 | Avraham Yaron |
| International Development Research Centre | 2412/15 | Avraham Yaron |
| Azrieli Foundation | 2412/15 | Avraham Yaron |
| Israel Science Foundation | 2412/15 | Avraham Yaron |

The funders had no role in study design, data collection and interpretation, or the decision to submit the work for publication.

### Author contributions

Hadas Sar Shalom, Conceptualization, Data curation, Formal analysis, Investigation, Writing—original draft, Writing—review and editing; Ron Goldner, Yarden Golan-Vaishenker, Data curation, Writing—original draft, Writing—review and editing; Avraham Yaron, Conceptualization, Resources, Supervision, Funding acquisition, Writing—original draft, Writing—review and editing

### Author ORCIDs

Avraham Yaron http://orcid.org/0000-0001-9340-7245

### Ethics

Animal experimentation: All of the animals were handled according to approved institutional animal care and use committee (IACUC) protocols (#06591013-2) of the Weizmann Institute of Science. All procedures employed on experimental animals, including their transportation, routine care and use in experiments are conducted in accordance with the Israel animal welfare law and guidelines, NIH guidelines and the Animal Welfare Act, the ethical standards and guidelines of FP7, with the EU directive 86/609/EEC as well as the revised directive 2010/63/EU on the protection of animals used for scientific purposes.

### Decision letter and Author response

Decision letter https://doi.org/10.7554/eLife.41162.027
Author response https://doi.org/10.7554/eLife.41162.028

## Additional files

### Supplementary files

• Transparent reporting form
DOI: https://doi.org/10.7554/eLife.41162.025

### Data availability

All data generated or analysed during this study are included in the manuscript. Source data files are provided.

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
