## [Decision Letter]

Thank you for submitting your article "Balance between BDNF and Semaphorins gates the sexually dimorphic innervation of the mammary gland" for consideration by *eLife*. Your article has been reviewed by three peer reviewers, one of whom is a member of our Board of Reviewing Editors, and the evaluation has been overseen by K VijayRaghavan as the Senior Editor. The reviewers have opted to remain anonymous.

The reviewers have discussed the reviews with one another and the Reviewing Editor has drafted this decision to help you prepare a revised submission.

Summary:

This study examines the roles of Semaphorins and PlexinA4 in sensory innervation of the mammary gland during early embryonic development. The findings support the idea that Sema6A (and possibly Sema3D) expressed in early (E12/13) mammary gland epithelial cells restricts innervation of the gland by TrkB positive sensory axons in females starting at E13. The phenotype of PlexinA4 mutants is more dramatic than Sema6A mutants, suggesting that signaling from additional Semaphorins, possibly Sema3D, may contribute to restricting sensory innervation of the mammary gland during early development. Experiments using genetic deletion of both PlexinA4 and BDNF (and TrkB.T1) suggest that a balance of BDNF and Semaphorin/PlexinA4 signaling controls innervation. The phenotypes presented are strong, the genetics correctly performed, and the observations interesting. The findings support a model in which the balance of neurotrophic factor signals and Semaphorin repulsive signals control early, sexual-dimorphic mammary gland innervation.

Essential revisions:

1) A concern of all three reviewers is over-interpretation of the results with respect to signaling mechanisms. The opposing effects of BDNF and Semaphorin/PlexinA4 signaling are difficult to disentangle with the single readout of axon innervation during the E12-14 timeframe examined. To illustrate this point, in male PlexinA4 knockout mice, gland hyper-innervation is seen as early as E12 when TrkB.T1 is not expressed (Liu et al., 2012) and the PlexinA4 phenotype appears to be unaffected by the onset of TrkB.T1 expression at the late E13 stages. These findings make it difficult to appreciate how the effects of PlexinA4 intersect with the BDNF/TrkB.T1 sequestration mechanism to contribute to the overall axonal regression in males during development. The authors should address this concern by presenting multiple interpretations of their findings, in the Discussion and elsewhere, since the reviewers do not view the experiments as definitive with respect to cross-talk between BDNF and Semaphorin signaling, nor of an absolute pruning function by Semaphorin signaling (as opposed to simple constraint of axon extension).

2) Related to point 1 and interpretations of the findings, the implication in the title and text that Semaphorin-PlexinA4 signaling helps to regulate the sexually dimorphic innervation pattern in mammary glands may not be compelling. The findings presented show that Semaphorin/PlexinA4 signaling is inhibitory to axon growth in both females and males, which is consistent with its well-established role. Semaphorins are not expressed in a sexually dimorphic manner in mammary glands, and it is unclear if there are sex-specific differences in PlexinA4 expression. The effects of PlexinA4 deletion in restricting axon growth could hold for any innervated target tissue, regardless of sex. The title and text should be modified to address this concern.

3) Some of the experiments are not as definitive as presented, and certain caveats need to be discussed. Growth cone collapse is not the same as axon pruning, and it should be noted that in these assays both male and female sensory axons undergo significant loss by E14, even in the absence of Semaphorin signaling. Also, it is likely that only a very minor fraction of all TrkB positive DRG neurons at certain axial levels innervate the mammary gland – the majority of TrkB positive neurons innervate other targets. In addition, the authors have not ruled out cell death resulting from lack of target innervation as a cause for the "pruning" effects that are observed in this system. Further, no evidence is presented to indicate timing of the expression of Semaphorin/PlexinA4 signaling components to selectively prune these axons. These caveats should be noted and discussed.

4) Why focus on Sema6A when Sema3D is expressed at high levels and causes collapse of growth cones? Sema6A mutant mice do not phenocopy the PlexinA4 mutant mice. Do the authors believe that the PlexinA4 effects are mediated by Sema3D in male embryos? Are Neuropilins expressed in TrkB neurons that innervate the mammary gland? Analysis of innervation using a Sema3D mutant mouse would be ideal, although it is not a requirement for publication if the mice may not be available to the investigators.

---

## [Author Response]

Essential revisions:1) A concern of all three reviewers is over-interpretation of the results with respect to signaling mechanisms. The opposing effects of BDNF and Semaphorin/PlexinA4 signaling are difficult to disentangle with the single readout of axon innervation during the E12-14 timeframe examined. To illustrate this point, in male PlexinA4 knockout mice, gland hyper-innervation is seen as early as E12 when TrkB.T1 is not expressed (Liu et al., 2012) and the PlexinA4 phenotype appears to be unaffected by the onset of TrkB.T1 expression at the late E13 stages. These findings make it difficult to appreciate how the effects of PlexinA4 intersect with the BDNF/TrkB.T1 sequestration mechanism to contribute to the overall axonal regression in males during development. The authors should address this concern by presenting multiple interpretations of their findings, in the Discussion and elsewhere, since the reviewers do not view the experiments as definitive with respect to cross-talk between BDNF and Semaphorin signaling, nor of an absolute pruning function by Semaphorin signaling (as opposed to simple constraint of axon extension).

We appreciate the concern about over-interpretation of the results, mainly regarding the axonal regression in males during development. We have therefore toned-down our statements throughout the text regarding the interpretation of the results, mostly regarding the role of the Semaphorins-PlexinA4 in axonal pruning.

Moreover, we discussed alternative models and the limitations of our findings in the Discussion section.

Lastly, in the course of the review process we preformed additional experiments in order to strengthen the conclusions of the paper. Specifically, we now show that the additional axons innervating the gland in the PlexinA4 female and male KO embryos, are TrkB positive. Therefore, the phenotypes indeed represent an increase in the local innervation (new Figure 4—figure supplement1 and new Figure 5—figure supplement1).

2) Related to point 1 and interpretations of the findings, the implication in the title and text that Semaphorin-PlexinA4 signaling helps to regulate the sexually dimorphic innervation pattern in mammary glands may not be compelling. The findings presented show that Semaphorin/PlexinA4 signaling is inhibitory to axon growth in both females and males, which is consistent with its well-established role. Semaphorins are not expressed in a sexually dimorphic manner in mammary glands, and it is unclear if there are sex-specific differences in PlexinA4 expression. The effects of PlexinA4 deletion in restricting axon growth could hold for any innervated target tissue, regardless of sex. The title and text should be modified to address this concern.

We changed the title to “Balance between BDNF and Semaphorins gates the innervation of the mammary gland”. We believe this title accurately reflects the findings that are presented in the paper. Moreover, we now emphasize throughout the paper that we have not detected any sexually dimorphic expression of the Semaphorins or PlexinA4.

3) Some of the experiments are not as definitive as presented, and certain caveats need to be discussed. Growth cone collapse is not the same as axon pruning, and it should be noted that in these assays both male and female sensory axons undergo significant loss by E14, even in the absence of Semaphorin signaling. Also, it is likely that only a very minor fraction of all TrkB positive DRG neurons at certain axial levels innervate the mammary gland – the majority of TrkB positive neurons innervate other targets. In addition, the authors have not ruled out cell death resulting from lack of target innervation as a cause for the "pruning" effects that are observed in this system. Further, no evidence is presented to indicate timing of the expression of Semaphorin/PlexinA4 signaling components to selectively prune these axons. These caveats should be noted and discussed.

The role of cell death in this pruning process was ruled out in the first paper that described the innervation of the mammary gland, as the axons are eliminated in the BAX KO (Liu et al., 2012). We now refer to this finding in the second paragraph of the Introduction. We agree that we do not show any evidence for temporal expression of Semaphorin/PlexinA4 signaling components to selectively prune these axons, and we now emphasize this throughout the text.

Importantly, our model suggests that such temporal control is *not* required, since dimorphic innervation is achieved by combining temporal control of BDNF signaling with constant PlexinA4 signaling.

Regarding the growth cone collapse assay: indeed, this assay was mainly used as a proxy for growth inhibition and repulsion. However, a recent study suggested that growth cone collapse is early and critical step in axonal degeneration in response to trophic deprivation (Unsain et al., 2018). Therefore, the hypersensitivity to PlexinA4 induced growth cone collapse may reflect promotion of the pruning process. We now refer to this point in the sixth paragraph of the Discussion.

4) Why focus on Sema6A when Sema3D is expressed at high levels and causes collapse of growth cones? Sema6A mutant mice do not phenocopy the PlexinA4 mutant mice. Do the authors believe that the PlexinA4 effects are mediated by Sema3D in male embryos? Are Neuropilins expressed in TrkB neurons that innervate the mammary gland? Analysis of innervation using a Sema3D mutant mouse would be ideal, although it is not a requirement for publication if the mice may not be available to the investigators.

We focused on Sema6A on the basis of its expression pattern, as visualized by the X-gal staining. Our analysis of the Sema6A KO and PlexinA4 KO strongly suggests that a number of additional Semaphorins might be involved in the control of the innervation. Hence, although Sema3D is a very good candidate, we cannot exclude the role of additional Semaphorins. Since we do not currently have access to a Sema3D mutant mouse, we respectfully submit that analyses of this and additional family members are beyond the scope of the present study, as noted also by the reviewer. Regarding the expression of Neuropilin-1, based on the fact that these neurons respond to Sema3D in vitro it is very likely that they express this receptor.